# Beneficial Effects of O-GlcNAc Stimulation in a Young Rat Model of Sepsis: Beyond Modulation of Gene Expression

**DOI:** 10.3390/ijms23126430

**Published:** 2022-06-09

**Authors:** Thomas Dupas, Antoine Persello, Angélique Blangy-Letheule, Manon Denis, Angélique Erraud, Virginie Aillerie, Aurélia A. Leroux, Matthieu Rivière, Jacques Lebreton, Arnaud Tessier, Bertrand Rozec, Benjamin Lauzier

**Affiliations:** 1Nantes Université, CHU Nantes, CNRS, INSERM, l’institut du thorax, F-44000 Nantes, France; thomas.dupas@univ-nantes.fr (T.D.); antoine.persello@gmail.com (A.P.); angelique.blangy--letheule@univ-nantes.fr (A.B.-L.); manon.denis@umontreal.ca (M.D.); angelique.erraud@univ-nantes.fr (A.E.); virginie.aillerie@univ-nantes.fr (V.A.); aurelia.leroux@oniris-nantes.fr (A.A.L.); bertrand.rozec@univ-nantes.fr (B.R.); 2Nantes Université, CNRS, CEISAM, UMR 6230, F-44000 Nantes, France; matthieu.riviere@univ-nantes.fr (M.R.); jacques.lebreton@univ-nantes.fr (J.L.); arnaud.tessier@univ-nantes.fr (A.T.)

**Keywords:** O-GlcNAcylation, post-translational modification, gene expression, transcriptomic, sepsis, therapeutic strategy

## Abstract

The young population, which is particularly at risk of sepsis, is, paradoxically, rarely studied. Acute stimulation of O-GlcNAcylation, a post-translational modification involved in metabolic regulation, cell survival and stress response, is beneficial in young rats with sepsis. Considering that sepsis impacts the gene expression profile and that O-GlcNAcylation is a regulator of transcription, the aims of this study are to (i) unveil beneficial mechanisms of O-GlcNAcylation and (ii) decipher the relationship between O-GlcNAcylation and transcription during sepsis. Endotoxemic challenge was induced in 28-day-old male rats using a lipopolysaccharide injection (*E. coli* O111:B4, 20 mg·kg^−1^) and compared to control rats (NaCl 0.9%). One hour after, rats were assigned to no therapy or fluidotherapy (NaCl 0.9%, 10 mL.kg^−1^) ± NButGT (10 mg·kg^−1^) to stimulate O-GlcNAc levels. Cardiac O-GlcNAcylation levels were evaluated via Western blot and gene transcription using 3′ SRP analysis. Lipopolysaccharide injection favorizes inflammatory state with the overexpression of genes involved in the NF-κB, JAK/STAT and MAPK pathways. NButGT treatment increased cardiac O-GlcNAcylation levels (*p* < 0.05). Yet, the mRNA expression was not impacted two hours after fluidotherapy or NButGT treatment. In conclusion, O-GlcNAc stimulation-induced beneficial effects are not dependent on the gene expression profile at the early phase of sepsis.

## 1. Introduction

Sepsis, qualified as a global priority by the World Health Organization, is a life-threatening medical condition characterized by organ dysfunction resulting from a dysregulated host response to infection [1]. In 2017, globally, 48.9 million people were affected by sepsis, of which 11 million died [2]. During sepsis, aberrant transcriptional control of gene expression is widely described [3,4] and may be related to post-translational modifications (PTMs) such as acetylation or methylation [5,6]. O-linked-N-acetyl glucosaminylation is a ubiquitous, well-conserved emerging PTM that is more commonly known as O-GlcNAcylation (O-GlcNAc), consisting of the addition of a monosaccharide (β-D-N-acetylglucosamine) to serine and threonine residues. In 2021, in humans, it was estimated that over 5000 cytosolic, nuclear and mitochondrial proteins are O-GlcNAcylated [7]. This modification is regulated by a single pair of enzymes, O-GlcNAc transferase (OGT) and O-GlcNAcase (OGA), which add and remove the GlcNAc moiety, respectively [8]. The O-GlcNAc cycle acts as a metabolic and stress sensor that regulates many of the key processes impaired during sepsis, including the production of inflammatory mediators and vascular contractility. O-GlcNAc has been found on proteins involved in all steps of transcription: the basal transcription machinery, epigenetic, chromatin remodeling complexes, and transcriptional cofactors [9,10]. In 2014, over 25% of the described O-GlcNAc-modified proteins are involved in transcriptional regulation [11].

Pharmacological elevations of O-GlcNAc have been shown to promote a survival signaling program in cells and tissues, favoring survival in models of heat stress, hypoxia-reoxygenation, oxidative stress, traumatic hemorrhage and ischemia-reperfusion injury of the heart [12]. Ferron and collaborators demonstrated that acute pharmacological stimulation of O-GlcNAc is protective in a lipopolysaccharide (LPS) and cecal ligation and puncture (CLP) adult rat model [13]. Furthermore, a study focusing on the impact of the treatment of septic shock in young animals highlighted that, despite higher basal levels of O-GlcNAcylation in young rats, increasing these levels is beneficial in this population too [14].

Although it is now understood that O-GlcNAcylation is a metabolic sensor that plays a key role in gene transcription, there is a lack of knowledge about the relationship between beneficial effects associated with the stimulation of O-GlcNAc levels and the regulation of gene expression in acute situation. In this context, this study proposes to examine the relationship between O-GlcNAc levels and the regulation of gene expression in young septic rats. A transcriptomic study was realized in a rat model of endotoxemic shock treated or not treated with a pharmacological inhibitor of OGA.

## 2. Results

### 2.1. Impact of Acute O-GlcNAc Stimulation on Young Septic Rats

Our model of septic shock has been studied previously and depicted a decrease in the mean arterial pressure and an increase in lactatemia and mortality in young rats. The administration of 10 mg/kg of NButGT (1,2-dideoxy-2′-propyl-α-d-glucopyranoso-[2,1-d]-Δ2′-thiazoline) [15] improves both mean arterial pressure and the median time of survival compared to rats treated with fluidotherapy (LPS + R) [14]. Although the O-GlcNAc stimulation is beneficial in young rats with septic shock, the mechanisms by which the O-GlcNAcylation acts remain unknown.

### 2.2. NButGT Treatment Increases Cardiac O-GlcNAc Levels without Impacting O-GlcNAc Cycling Enzymes

The cardiac O-GlcNAc levels were not modified between the CTRL, LPS and LPS + R group (CTRL: 1.00 ± 0.05; LPS: 0.96 ± 0.06; LPS + R: 1.12 ± 0.06) (Figure 1A). Pharmacological inhibition of OGA with NButGT increases cardiac O-GlcNAc levels compared to the LPS + R group (NButGT: 1.84 ± 0.09; *p* < 0.05) (Figure 1A). Interestingly, neither ncOGT (nucleocytoplasmic OGT) nor lOGA (long OGA) cardiac expression were modified in the studied groups, thus validating that NButGT increases O-GlcNAc cardiac levels without impacting the associated enzymes (Figure 1B,C).

### 2.3. The LPS Injection Stimulates Pro-Inflammatory Pathways While O-GlcNAc Stimulation Does Not Impact Gene Expression Profiles

Due to the existing and well-described link between O-GlcNAcylation and transcription, a 3’ SRP study was performed to decipher the role of the O-GlcNAcylation on the gene expression profile in young septic rats.

Heatmap and principal component analysis (PCA) showed clear separation of the groups and individuals in response to endotoxemia in healthy rats without separation compared to rats treated with either fluidotherapy or NButGT (Figure 2 and Appendix A).

After 3′SRP analysis of heart transcriptome, a total of 1483 genes of the 13730 detected genes matched our selection criteria (Log2 (fold change) <−1 or >1 and p-adjusted <0.05). Differentially expressed genes’ (DEGs) visualization with heatmap showed a good clusterization of samples within groups (Figure 3A). Gene clusterization showed two main clusters of DEGs in either the CTRL or LPS groups. These DEGs were retrieved in the volcano plot representation that showed 828 (55%) genes were overexpressed in the LPS group, while 655 (45%) genes were overexpressed in the CTRL group (Figure 3B).

An analysis of over-represented gene ontologies (GO) biological processes (BP) was performed to decipher the impact of sepsis-induced modification on transcriptome in the heart. Biological processes in which DEGs participate the most were identified by enrichment analysis using the “clusterProfiler” R package. Overall, 1490 GO terms were identified among which 1450 GO terms for genes were overexpressed in the LPS group and 40 GO terms for genes overexpressed in the CTRL group.

GO terms for genes overexpressed in the CTRL group are related to BP associated with the cell division cycle such as “mitotic nuclear division” or “sister chromatid segregation”. In addition, biological processes related to the regulation of actin filament cytoskeleton (“actin filament bundle organization”; “regulation of stress fiber assembly”) and heart development (“endocardial cushion development”; “atrioventricular valve development”) are found. All these data confirm that, at 28 days of age, the rats’ hearts are still in a development or maturation phase (Figure 3C).

As expected, most of the over-represented BP in the endotoxemic shock group are related to inflammation, cytokine production (“positive regulation of cytokine production”; “cytokine-mediated signaling pathway”; “response to tumor necrosis factor”) and leukocyte activation (“regulation of cell-cell adhesion”; “leukocyte mediated immunity”; “regulation of lymphocyte activation”). In addition, BP such as “cellular response to lipopolysaccharide” and “regulation of NF-kappaB signaling” were identified among the overexpressed genes in sepsis rats. The top fifteen pathways enriched according to clusterProfiler demonstrated that genes involved in inflammation and the response to the lipopolysaccharide, two major pathways in the pathophysiology of sepsis, are overexpressed in the LPS group.

In addition, a GO-BP enrichment of clusters comparison showed a predominance of LPS related annotation in this study (Figure 3D).

An analysis of pathways using the KEGG database (released from 1 April 2022) for *Rattus norvegicus* species (“*rno*”), thanks to KEGG enrichment of DEGs and pathways visualization (Figure 4), showed that three main pathways were altered in the heart 3 h after LPS injection: the MAPK signaling pathway (rno04010), the JAK-STAT signaling pathway (rno04630) and the NF-κB signaling pathway (rno04064). Interestingly, 89% (17/19), 77% (24/31) and 95% (39/41) of differentially expressed genes in the JAK-STAT, the NF-κB, and the MAPK signaling pathway, respectively, are overexpressed in the LPS group. All these data demonstrate a pro-inflammatory and adaptative response to the lipopolysaccharide injection.

Fluidotherapy does not impact gene expression profiles. Surprisingly, despite its involvement in the transcription process, O-GlcNAcylation does not impact gene expression profiling either (Figure 5).

## 3. Discussion

Sepsis is defined as a dysregulated response of the host to an infectious pathogen, which is responsible for 1 in 5 deaths (11 million) per year worldwide [2]. This response is the result of a systemic inflammatory response syndrome (SIRS) following an infection. It combines the early activation of pro- and anti-inflammatory responses with non-immunological cardiovascular, neuro-hormonal, autonomic, bioenergetic, metabolic, and coagulation changes [16,17,18,19].

### 3.1. Lipopolysaccharide Injection Promotes Pro-Inflammatory State

The inflammatory context during sepsis is highly extensive and complex so that it is described in the literature as a “cytokine storm”, composed of both anti- and pro-inflammatory cytokines. Pro-inflammatory molecules (e.g., Interleukin-18, TNF, Interleukin-6, Interleukin-17, Interleukin-1β) are predominantly represented in the acute phase and linked to activity maintenance of signaling pathways (e.g., MAPK, mTOR, NF-κB, JAK-STAT3) and lead to multi-organ dysfunction [20]. As expected, the lipopolysaccharide injection promotes the gene expression of pro-inflammatory cytokines as *Il-18*, *Il-6, Il-17, Il-1β* and *Tnf-*α in a rat’s heart, thus validating a strong and quick inflammatory response. The inducible transcription factor NF-κB is the major regulator of the inflammatory response during sepsis and acts as a mediator of pro-inflammatory gene induction but also in multi-organ dysfunction [21,22,23,24]. As previously described by Chen et al. in a sepsis mouse model [25], we demonstrated that 3 h after injection, lipopolysaccharides promote inflammatory pathways in the hearts of young rats with the increase in genes involved in the regulation of NF-κB signaling. Reinforcing this observation, genes involved in the MAPK, mTOR, TLR2/MyD88 and JAK/STAT pathway are also up-regulated in the LPS group, as recently described in sepsis patients [26], confirming the pertinence of such approaches. JAK/STAT3 is a major pathway described as cardio-protective [27,28]; however, this pathway could also contribute to cardiac dysfunction [29,30,31], notably during septic shock [32,33]. Recent studies provide some evidence of therapeutic potential for targeting STAT3 in the cecal and ligature-puncture-induced sepsis model [34,35]. Moreover, *Stat5a* and *Rela*, two genes overexpressed in the LPS group, have recently been described as crucial for prognosis in septic patients [36]. Although compensatory anti-inflammatory response syndrome (CARS) is largely described for sepsis, we report a poor presence of DEGs implicated in the anti-inflammatory response, notably due to the early phase of acute septic shock (3 h between shock induction and gene expression profile analysis). Pro-inflammatory cytokines allow the recruitment of inflammatory leukocytes and the upregulation of adhesion molecules. In our study, we demonstrated that gene encoding for adhesion molecules expressed in leukocytes (*Sele* for E-Selectin and *Selp* for P-Selectin) and on the endothelium (*Icam1* and *Vcam1*) are overexpressed in the LPS group. This result is in accordance with several studies that have demonstrated that adhesion molecules are increased in children with septic shock and can be related to the clinical severity of the sepsis [37,38,39], although aging must be considered [40].

### 3.2. Gene Expressions Involved in the Integrated Stress Response and Cell Death Are Upregulated with Sepsis

The integrated stress response (ISR) is a well-conserved pathway that is activated in response to physiological changes and pathological conditions, which can lead to cell death if homeostasis is not restored [41]. Extrinsic stressors such as hypoxia or viral infection can activate this pathway. However, the accumulation of unfolded proteins in the endoplasmic reticulum (ER) can also activate the ISR. According to the results described in the literature [42,43,44], we demonstrated that the LPS injection promotes the expression of ER stress-related genes by activating transcription factors 6 and 4 (ATF6 and ATF4), which promotes the transcription of genes involved in the ER quality control and apoptosis, respectively.

### 3.3. Transcriptomic Pattern throughout Aging and Species

The modulation of the gene expression profile during sepsis is largely described in adults. However, the young population had pathophysiological characteristics and is particularly vulnerable to septic shock [45,46]. Thus, evaluating the transcriptomic response in the young is an important issue to (i) improve knowledge about pathophysiological mechanisms and (ii) adapt therapeutic care. Studies that have looked at variations in gene expression in young people with septic shock are not unanimous with variations [47] or not [48] throughout development. As in our rat model, recent studies agreed on a modification of the gene expression profile during sepsis in the young. The DNA Damage Inducible Transcript 4 (*Ddit4*), a gene associated with higher risks of mortality, has notably been identified as overexpressed in the young with septic shock [49,50,51]. Animal septic shock models have been largely used over the past decades to better understand pathophysiological mechanisms. Interestingly, several studies observed a similar modification of the gene expression profiles between the sepsis model and clinical samples, suggesting that animal models are relevant to evaluate the molecular mechanisms involved in the pathophysiology of sepsis [52,53,54], although this may be discussed as to their relevance to the clinic [55,56,57].

### 3.4. Beneficial Effects of O-GlcNAcylation Stimulation in Sepsis: New Insights

As described in previous studies, O-GlcNAc stimulation is beneficial in septic shock in both adults and pup rats [13,14]. Yet, the mechanisms associated with the protective effects are not completely elucidated and remain to be understood. Despite the known involvement of O-GlcNAcylation in the transcription regulation process, we do not report in our study any impact of NButGT on the gene expression profile 3 h after shock induction. Considering that O-GlcNAcylome is impacted by NButGT treatment and had no impact on transcriptome over such a period of time, one should consider and evaluate the impact of O-GlcNAcylation on key pathways identified in our O-GlcNAcylomic study such as inflammation, integrated stress response or cell death/survival pathways to unveil the beneficial mechanisms induced by O-GlcNAcylation. Additionally, O-GlcNAc can compete with phosphorylation, so it could be of interest to evaluate the phosphorylome under our conditions [58].

## 4. Methods

### 4.1. Reagents

The O-GlcNAcase inhibitor NButGT was synthesized using Matthew S. Macauley methods [59].

### 4.2. Animal Models

Male Wistar Han rats (Charles River, Saint-Germain-Nuelles, France) were housed under standard conditions of temperature (21–24 °C), humidity (40–60%) and a 12 h light/dark cycle with the light period starting at 07:00 a.m. Food and water were available ad libitum. Endotoxemic shock was induced in 28-days-old Wistar Han rats and compared to control rats (injection of NaCl 0.9%—CTRL) as previously described [14]. One hour after LPS injection, the rats were randomly assigned to no therapy (LPS) or fluidotherapy (NaCl 0.9%, 10 mL/kg—LPS + R) ± NButGT (NButGT, 10 mg/kg—NButGT) to increase O-GlcNAcylation levels (Figure 6).

### 4.3. Heart Sampling

Rats were anesthetized by the inhalation of an isoflurane/O_2_ mixture (Forène, Abbott, Rungis, France) (induction: 5% isoflurane, flow rate: 1 L/min; maintenance: 2% isoflurane, flow rate: 0.5 L/min). The beating heart was freeze-clamped using a Wollenberger clamp to preserve post-translational modifications for biochemical and molecular biology analyses under standardized conditions (between 8:30 a.m and 10:00 a.m) to avoid circadian modifications on protein O-GlcNAcylation levels.

### 4.4. Tissue Preparation

Frozen hearts were crushed to *obtain* powder as previously described [60]. Briefly, frozen hearts were crushed in liquid nitrogen using a mortar to obtain a homogenous powder that was used for protein or mRNA analyses. In order to preserve post-translational modifications, all steps were carried out in liquid nitrogen. The powder was then stored at −80 °C.

### 4.5. Protein Extraction and Western Blots

Western blotting experiments were performed as previously described [14]. Analysis was performed using Image Lab software (Image Lab 6.1, Bio-Rad, CA, USA).

### 4.6. RNA Extraction, Quality Evaluation and DGESeq Processing and Analyses

RNA extraction from deep-frozen samples was processed according to Dupas and collaborators [60] and RNA was recovered in sterile RNAse-free water (Qiagen, France), dosed with a NanoDrop ND-1000 Spectrophotometer (Thermoscientific, France), and quality was controlled with RIN measures (Bioanalyzer, Agilent, United States). Retro-transcription was achieved using the cDNA Reverse Transcription High-Capacity kit (Applied Biosystem, Courtaboeuf, France). Sample preparation, the verification of sample quality and quantity, and transcriptomics were performed according to the protocol explained and detailed by Charpentier et al. [61]. The identification and selection of differentially expressed genes was performed with the R package “DESeq2”, and “clusterProfiler” was used for enrichment analyses and associated graphical representation [62]. Gene ontology (GO), biological process and KEGG (Kyoto Encyclopedia of Genes and Genomes) enrichments to highlight the most statistically represented annotations of biological processes and signaling pathways among the selected genes GO and KEGG enrichment analyses were performed. Genes are considered differentially expressed if Log2 fold change <−1 or >1 and the Benjamini–Hochberg adjusted *p* value < 0.05.

### 4.7. Statistical Analyses

The results were expressed as an average ± SEM of *n* different rats. Analyses of Western blots were expressed in relation to the average of the stain-free samples and then normalized to the average of the control samples (CTRL). For CTRL-LPS-LPS + R-NButGT Western blots, data were analyzed using a Kruskal–Wallis test followed by an uncorrected Dunn test. A value of *p* < 0.05 was considered significant. All statistical calculations and graphs (except those performed with R software) were performed using GraphPad Prism software (version 8.4.2).

## 5. Conclusions

Our study demonstrated for the first time that beneficial effects associated with the O-GlcNAc stimulation in young rats with sepsis are independent of the gene expression modulation in the early phase. Identifying the mechanisms involved in beneficial effects remains a major challenge to decipher O-GlcNAcylation in septic shock. We also demonstrated that, depending on the context, O-GlcNAcylation does not always have an impact on gene expression, suggesting that the relationship is much more complicated than it seems, and this will prove to be a major issue in the years to come.

## Figures and Tables

**Figure 1 ijms-23-06430-f001:**
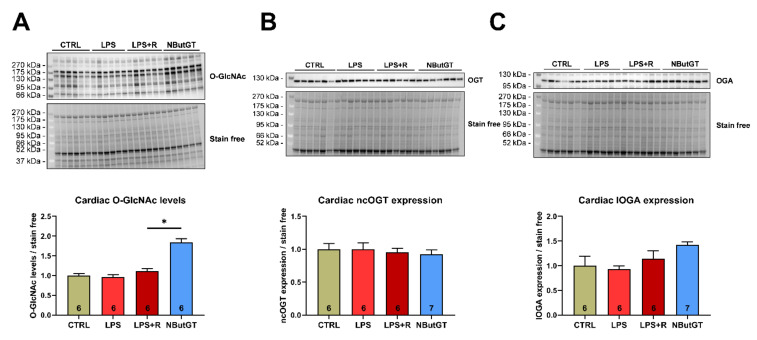
Impact of septic shock and NButGT on O-GlcNAcylation. Evaluation using Western blot of the O-GlcNAcylation levels (**A**), ncOGT (**B**), and lOGA (**C**) expression in the heart in CTRL, LPS, LPS + R and NButGT group. Statistical significance was assessed using a Kruskal–Wallis test with uncorrected Dunn’s post-test. Quantification was performed in relation to stain-free. Results are expressed as mean ± SEM. CTRL: control group; LPS: i.v. injection of LPS (20 mg/kg); LPS + R: s.c. administration of 10 mL/kg of NaCl 0.9%; NButGT: fluidotherapy supplemented with NButGT (10 mg/kg). *n* = 6–7. *: *p* < 0.05.

**Figure 2 ijms-23-06430-f002:**
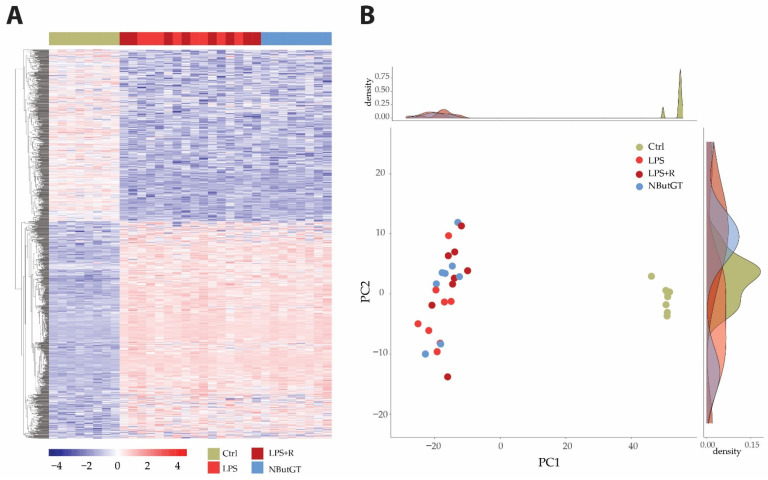
Heatmap and principal component analysis. Transcriptome analyses were made from whole heart of rats injected with saline (CTRL), LPS (LPS) treated with either fluidotherapy (LPS + R) or NButGT (NButGT), and this allowed the identification of 13730 genes. Differentially expressed genes (DEGs) were selected by their fold change of expression between 2 conditions log2Ratio(Fold change)>1 or log2Ratio(Fold change)>−1 and an adjusted *p* value < 0.05 using the R package “Deseq2”. Heatmap visualization (**A**) of the 1483 DEGs with the comparison Ctrl vs. LPS and the 1 DEG with the comparison LPS vs. LPS + R showed that Ctrl and NButGT rats were well clustered but LPS and LPS + R showed no clusterization. Plotting of individuals and frequencies from Principal Component Analysis (PCA) of DGEseq data using R statistical language showed Ctrl rats clustered but no separation of LPS, LPS + R and NButGT individuals (**B**).

**Figure 3 ijms-23-06430-f003:**
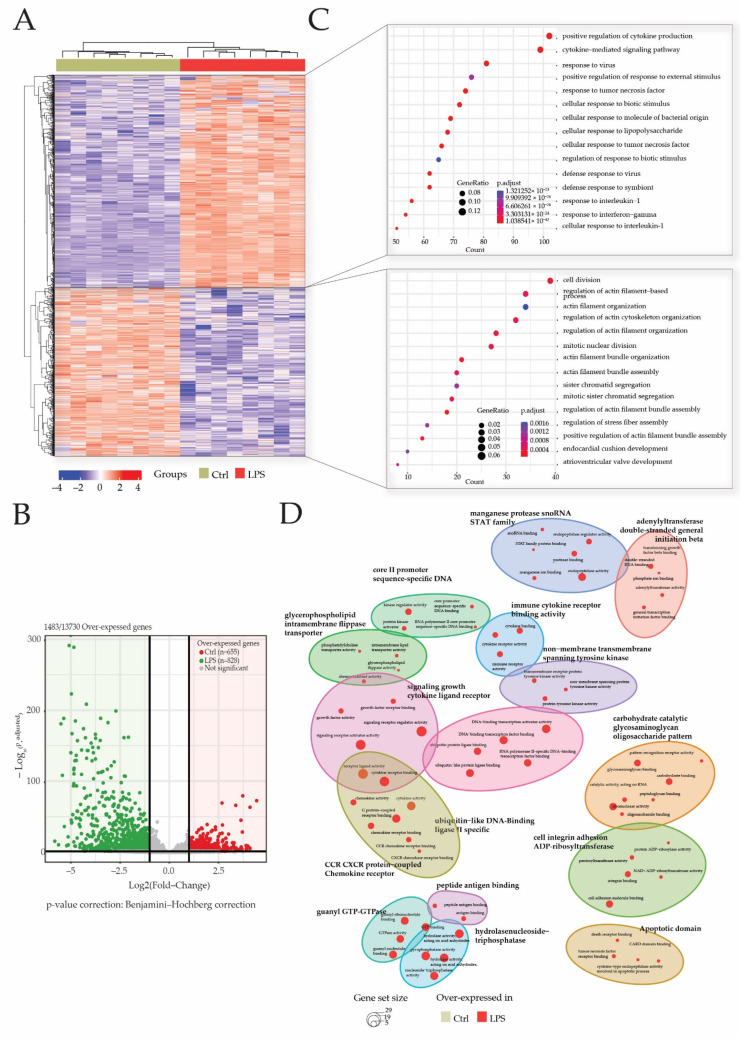
LPS injection promotes cytokine and inflammatory pathways. Transcriptome analyses were made from whole heart of rats injected with LPS (LPS) or saline (CTRL), and this allowed the identification of 13730 genes. Differentially expressed genes (DEGs) were selected by their fold change of expression (log2Ratio(LPS/CTRL) >1 or (log2Ratio(LPS/CTRL) >−1 and an adjusted *p* value < 0.05 using the R package “Deseq2”. Heatmap visualization (**A**) of the 1483 DEGs showed that groups were well clustered and revealed two main clusters of overexpressed genes, respectively, in the LPS group and CTRL group. Log_10_ (p.adjusted) and log2FoldChange of these DEGs are represented in the volcano plot (**B**). Gene ontology (GO) enrichments were conducted to decipher biological processes in which DEGs were mostly involved in both clusters (**C**) and within the comparison of both clustered (**D**).

**Figure 4 ijms-23-06430-f004:**
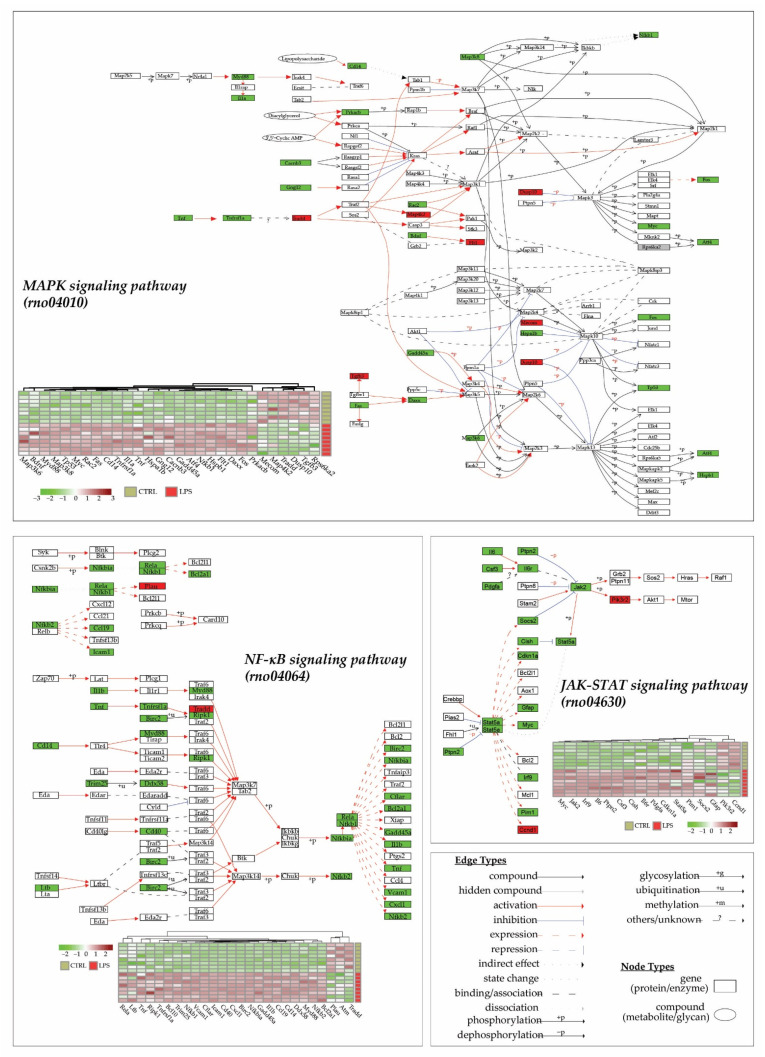
LPS injection promotes cytokine and inflammatory pathways. Transcriptome analyses were made from whole heart of rats injected with LPS (LPS) or saline (CTRL), and this allowed the identification of 13730 genes. Differentially expressed genes (DEGs) were selected by their fold change of expression (log2Ratio(LPS/CTRL) >1 or (log2Ratio(LPS/CTRL) >−1 and an adjusted *p* value < 0.05 using the R package “Deseq2”. KEGG (Kyoto Encyclopedia of Genes and Genomes) enrichments were conducted to decipher most impacted pathways. Represented here are 4 main pathways in the heart that were impacted by LPS injection in CTRL rats: the MAPK signaling pathway (rno04010), the JAK-STAT signaling pathway (rno04630), and the NF-κB signaling pathway (rno04064). Each DEGs expression in pathways is represented in a heatmap.

**Figure 5 ijms-23-06430-f005:**
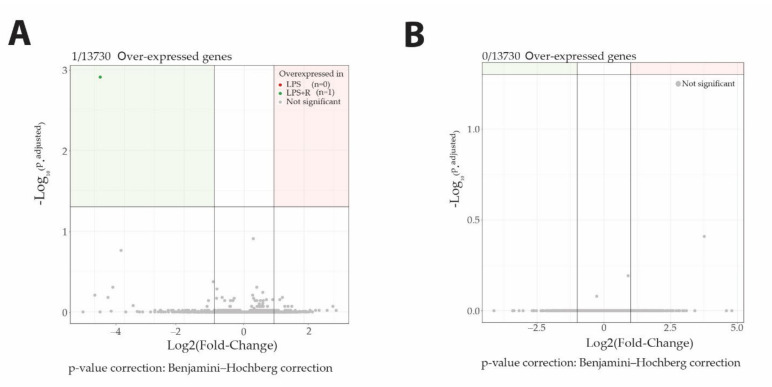
Neither fluidotherapy nor NButGT treatments affected gene transcription after LPS injection in rats. Volcano plot representation of log2FoldChange calculation and statistical differences with Log_10_ (p.adjusted) of LPS vs. LPS + R comparison (**A**) and LPS + R vs. NButGT comparison (**B**). Genes were considered differentially expressed when matching the following criteria: log2FoldChange >1 or log2FoldChange <−1 and Significance (p.adjusted) <0.05.

**Figure 6 ijms-23-06430-f006:**
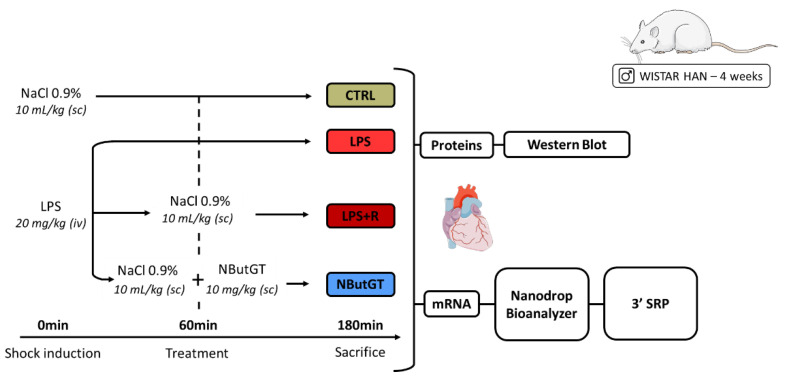
Model of endotoxemic shock in the young rat and analyses performed. CTRL: control group; LPS: i.v., injection of LPS (20 mg/kg); LPS + R: s.c. administration of 10 mL/kg of NaCl 0.9%; NButGT: fluidotherapy supplemented with NButGT (10 mg/kg).

## Data Availability

The data that support the findings of this study are available from the corresponding author upon reasonable request.

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
