# Peer review of "Beneficial Effects of O-GlcNAc Stimulation in a Young Rat Model of Sepsis: Beyond Modulation of Gene Expression"

_ijms, 2022, doi:10.3390/ijms23126430_

Round 1

Reviewer 1 Report

The work presented by Thomas Dupas titled: “Beneficial effects of O-GlcNAc stimulation in a young rat model of sepsis: beyond modulation of gene expression” is well written although some points are not clear enough. The topic is interesting and therefore, it adds new information to the subject area of young sepsis development and treatment which is a cutting edge area particularly translating this base concept into the management in the future of premature newborn which develops sepsis in the first 3 days of their life.

Say that, here after my comment:

line 53 defines the acronymous LPS and LPC

line 67 “This study suggests that acute O-GlcNAc beneficial effects are not mediated by modulation of gene expression.” Do not write this sentence in the introduction section it might be better in the conclusions.

line 73 define the acronymous NButGT (it is the OGA inhibitor 1,2-dideoxy-2′-propyl-α-d-glucopyranoso-[2,1-d]-Δ2′-thiazoline), please refer to 10.1016/j.chembiol.2010.07.006

since the LPS+R group is defined in the material and method section at the end of the manuscript it will be better to clarify it also in the results

in figure 2 left underneath the heatmap, only the expression levels colored insert, put up the caption related to crtl, and treated groups

please adds at line 164 KEGG database before defining rno###

harmonize the discussion section by taking off the paragraphs

line 294 heart is a tissue really hard to manage please detail better the procedure to obtain powder

Reviewer 2 Report

The studies described in the paper deal with the continuation of  a novel area of the research recently initiated by two groups, the authors of this paper (M. Ferron and colleagues, 2019; M. Denis and colleagues., 2021) and J.-S. Hwang and colleagues (2021). Both studies have strongly demonstrated the protective effect of increased  O-GlcNAcylation on sepsis, however, mechanisms of the protection have remained unclarified. Presented study  step-by-step  has attempted, firstly,  to confirm the protection from sepsis by chemically-driven increased  O-GlcNAcylation in a young rat sepsis model using protocol mimicking the clinically relevant condition (fluidotherapy and NButGT treatment). Secondly,  it determined if increasing the candidate genes expression  is responsible for the anti-sepsis effect. The results confirmed using a modified sepsis treatment model the previous effect, however, did not reveal any mechanism. Furthermore, in a rat sepsis model, the study has clearly demonstrated no contribution of various candidate genes encoding NF-κB, JAK/STAT and MAPK pathways to abrogation of the LPS-induced sepsis course induced by increased  O-GlcNAcylation. The strong evidence was obtained that although treatment with the O-GlcNAcylation inducer NButGT caused cardiac O-GlcNAcylation and blockade of sepsis development, no alterations in gene transcription were noticed.  The advantage of the paper is that it clearly shows that the sepsis blockade by increased O-GlcNAcylation is not associated with changes in transcription of the most pathogenically related genes enhanced during LPS induction at least during early phase of experimental sepsis. However, the intrigue still remains because the mechanisms responsible for the anti-sepsis effect of O-GlcNAcylation have been nor clarified, nor hypothetically discussed.  The latter may represent some weakness of the paper since in contrast to previous papers by the authors, no any mechanistical hypothesis have been proposed for the non-genetic causes of repeatedly demonstrated, significant anti-sepsis effect of the NButGT treatment. The improved discussion section of the paper may help rise further the interest to studies the phenomenon that possesses strong translational potential in the area of developing strategies to treat sepsis.

Author Response

Response to reviewer

The authors thank the reviewer for agreeing to review the paper and for all their pertinent comments.

Reviewer 2

  1. The studies described in the paper deal with the continuation of  a novel area of the research recently initiated by two groups, the authors of this paper (M. Ferron and colleagues, 2019; M. Denis and colleagues., 2021) and J.-S. Hwang and colleagues (2021). Both studies have strongly demonstrated the protective effect of increased  O-GlcNAcylation on sepsis, however, mechanisms of the protection have remained unclarified. Presented study  step-by-step  has attempted, firstly,  to confirm the protection from sepsis by chemically-driven increased  O-GlcNAcylation in a young rat sepsis model using protocol mimicking the clinically relevant condition (fluidotherapy and NButGT treatment). Secondly,  it determined if increasing the candidate genes expression  is responsible for the anti-sepsis effect. The results confirmed using a modified sepsis treatment model the previous effect, however, did not reveal any mechanism. Furthermore, in a rat sepsis model, the study has clearly demonstrated no contribution of various candidate genes encoding NF-κB, JAK/STAT and MAPK pathways to abrogation of the LPS-induced sepsis course induced by increased  O-GlcNAcylation. The strong evidence was obtained that although treatment with the O-GlcNAcylation inducer NButGTcaused cardiac O-GlcNAcylation and blockade of sepsis development, no alterations in gene transcription were noticed.  The advantage of the paper is that it clearly shows that the sepsis blockade by increased O-GlcNAcylation is not associated with changes in transcription of the most pathogenically related genes enhanced during LPS induction at least during early phase of experimental sepsis. However, the intrigue still remains because the mechanisms responsible for the anti-sepsis effect of O-GlcNAcylation have been nor clarified, nor hypothetically discussed.  The latter may represent some weakness of the paper since in contrast to previous papers by the authors, no any mechanistical hypothesis have been proposed for the non-genetic causes of repeatedly demonstrated, significant anti-sepsis effect of the NButGT treatment. The improved discussion section of the paper may help rise further the interest to studies the phenomenon that possesses strong translational potential in the area of developing strategies to treat sepsis.

 The authors thank the reviewer for this comment. In accordance with the comment, a paragraph has been added in the discussion section as follows:

3.4. Beneficial effects of O-GlcNAcylation stimulation in sepsis: new insights

As described in previous studies, the O-GlcNAc stimulation is beneficial in septic shock in both adults and pups rats [1,2]. Yet, the mechanisms associated with the protective effects are not completely elucidated and remain to be understood. Despite the known involvement of the O-GlcNAcylation in the transcription regulation process; we do not report in our study any impact of the NButGT on the gene expression profile 3 hours after shock induction. Considering that O-GlcNAcylome is impacted by NButGT treatment and had no impact on transcriptome on such period of time, one should consider evaluate the impact of O-GlcNAcylation on key pathways identified in our O-GlcNAcylomic study such the inflammation, the integrated stress response or the cell death/survival pathways to unveil the beneficial mechanisms induced by the O-GlcNAcylation. Also, O-GlcNAc can compete with phosphorylation, it could be of interest to evaluate the phosphorylome under our condition [3].”

  1. Ferron, M.; Cadiet, J.; Persello, A.; Prat, V.; Denis, M.; Erraud, A.; Aillerie, V.; Mevel, M.; Bigot, E.; Chatham, J.C.; et al. O-GlcNAc Stimulation: A New Metabolic Approach to Treat Septic Shock. Sci Rep 2019, 9, 18751, doi:10.1038/s41598-019-55381-7.
  2. Denis, M.; Dupas, T.; Persello, A.; Dontaine, J.; Bultot, L.; Betus, C.; Pelé, T.; Dhot, J.; Erraud, A.; Maillard, A.; et al. An O-GlcNAcylomic Approach Reveals ACLY as a Potential Target in Sepsis in the Young Rat. IJMS 2021, 22, 9236, doi:10.3390/ijms22179236.
  3. van der Laarse, S.A.M.; Leney, A.C.; Heck, A.J.R. Crosstalk between Phosphorylation and O-GlcNAcylation: Friend or Foe. The FEBS Journal 2018, 285, 3152–3167, doi:10.1111/febs.14491.